# Overexpression of Both Human Sodium Iodide Symporter (NIS) and BRG1-Bromodomain Synergistically Enhances Radioiodine Sensitivity by Stabilizing p53 through NPM1 Expression

**DOI:** 10.3390/ijms24032761

**Published:** 2023-02-01

**Authors:** Juri Na, Chul-Hee Lee, June-Key Chung, Hyewon Youn

**Affiliations:** 1Department of Nuclear Medicine, Seoul National University College of Medicine, Seoul 03080, Republic of Korea; 2Cancer Research Institute, Seoul National University College of Medicine, Seoul 03080, Republic of Korea; 3Cancer Imaging Centre, Seoul National University Hospital, Seoul 03080, Republic of Korea

**Keywords:** sodium iodide symporter, radioiodine therapy, iodine-131, brahma-related gene 1 bromodomain, radiosensitization, thyroid cancer

## Abstract

Improved therapeutic strategies are required to minimize side effects associated with radioiodine gene therapy to avoid unnecessary damage to normal cells and radiation-induced secondary malignancies. We previously reported that codon-optimized sodium iodide symporter (oNIS) enhances absorption of I-131 and that the brahma-associated gene 1 bromodomain (BRG1-BRD) causes inefficient DNA damage repair after high-energy X-ray therapy. To increase the therapeutic effect without applying excessive radiation, we considered the combination of oNIS and BRG1-BRD as gene therapy for the most effective radioiodine treatment. The antitumor effect of I-131 with oNIS or oNIS+BRD expression was examined by tumor xenograft models along with functional assays at the cellular level. The synergistic effect of both BRG1-BRD and oNIS gene overexpression resulted in more DNA double-strand breaks and led to reduced cell proliferation/survival rates after I-131 treatment, which was mediated by the p53/p21 pathway. We found increased p53, p21, and nucleophosmin 1 (NPM1) in oNIS- and BRD-expressing cells following I-131 treatment, even though the remaining levels of citrulline and protein arginine deiminase 4 (PAD4) were unchanged at the protein level.

## 1. Introduction

Radioactive iodine therapy is a type of internal radiation treatment that introduces a radiation source into the body and is used specifically for thyroid diseases, including benign hyperthyroidism and thyroid cancer [1]. The most common iodine radioisotope used for treatment, iodine-131 (I-131), releases radiation during the decay process by beta-emission, with a maximal energy of 606 keV (90%). This can be used therapeutically, as it penetrates neighboring cells up to several millimeters away and causes apoptosis. I-131 also emits a gamma-ray with a maximal energy of 364 keV (10%), which is helpful for imaging-based diagnosis [2,3,4]. The success rate of radioactive iodine therapy is dependent on iodine uptake into cells through NIS. NIS co-transports two sodium ions and an iodine ion along an electrochemical gradient, which leads to the active accumulation of iodine [5]. Following the cloning of NIS cDNA [6,7], the NIS gene has been proposed to be a powerful reporter and therapeutic gene [8,9,10]. By transfecting the NIS gene into tumors, cancer cells can take up the required therapeutic level of radioiodine [11,12,13]. Previously, we reported that a codon-optimized human sodium iodide symporter (oNIS) enhances therapeutic efficacy by increasing the amount of iodine uptake [14], which could be beneficial for cancer patients with low NIS expression. Although increasing iodine uptake improves the therapeutic effect, I-131 therapy still has limitations, given the health risks associated with using an isotope, such as unnecessary damage to normal cells [15,16,17], which could lead to radiation-induced secondary malignancies and potential genetic alterations [18,19].

To improve the shortcomings of current I-131 treatments, we considered a novel gene therapy that involves the application of a radiosensitizer along with oNIS to maximize treatment efficacy [20] by minimizing the dose of I-131 to achieve an equivalent outcome. After radiotherapy, residual DNA double-strand breaks (DSBs), which are the result of inefficient DNA repair, are considered lethal in cancer [21,22]. In the eukaryotic genome, repair of DSBs involves both post-translational modifications of nucleosomes and accumulation of DNA repair proteins at the site of damage [23], which lead to histone modifications and chromatin remodeling; these are therefore important factors for DSB repair and DNA damage repair (DDR) [24,25]. Brahma-related gene 1 (BRG1) is an ATPase subunit of the SWI/SNF chromatin remodeling complex [26]. The bromodomain (BRD) of BRG1 binds to an acetylated histone tail [27], which then interacts with pS139 of γ-H2AX, resulting in DSB repair [28,29]. We previously reported that overexpression of BRG1-BRD in cancer cells leads to inefficient DNA repair after ionizing radiation (IR) with high-energy X-rays, which suggests that BRG1-BRD can be used as an efficient radiosensitizer [30]. Therefore, here, we evaluated the synergistic effect of I-131 treatment with dual oNIS and BRG1-BRD overexpression, which occurs via increased iodine uptake and inhibition of the DNA damage repair process. Furthermore, we investigated the pathway of the BRG1-BRD-induced DNA damage response through p53, p21, and a multifunctional chromatin-binding protein (nucleophosmin 1, NPM1) after I-131 treatment.

## 2. Results

### 2.1. Overexpression of oNIS and BRG1-BRD Enhances the Effect of I-131 by Increasing DNA DSB Levels

We previously identified the effect of BRG1-BRD overexpression in cells treated with external radiation by mitigating the crosstalk between the chromatin remodeling SWI/SNF complex and the acetylated lysine in histones [30]. In this study, we combined overexpression of oNIS and BRG1-BRD (oNIS+BRD) to enhance radiation sensitivity to I-131 treatment. Stable oNIS and oNIS+BRD overexpression were established in FRO cells by retroviral transduction, and BRG1-BRD overexpression at both the mRNA and protein levels was evaluated through myc-tag (Figure 1A,B and Appendix A). The BRG1-BRD-overexpressing cell line was selected by the monoclonal selection, and colony D10 (Appendix A) was chosen for further experiments. Expression levels of oNIS were consistent in both oNIS and oNIS+BRD cell lines, and the stable function of oNIS was shown directly through iodine uptake levels (Figure 1B,C). BRG1-BRD overexpression in oNIS cells did not affect iodine uptake, cell morphology, or the proliferation rate (Figure 1C–E).

We then investigated whether oNIS+BRD modulates the radiation sensitivity of cancer cells. The number of colonies decreased with I-131 treatment in a dose-dependent manner. The cell survival rate after I-131 (2.22 MBq) treatment for nontransfected control, oNIS-, and oNIS+BRD-expressing cells was 65%, 29%, and 12%, respectively (*p* < 0.05) (Figure 2A,B). To determine whether increased radioiodine sensitivity of cancer cells expressing oNIS or oNIS+BRD was associated with defective DSB repair, we measured the level of DNA DSBs by counting the number of γ-H2AX foci generated after damage. The number of γ-H2AX foci was 1.5-fold higher in oNIS+BRD cells compared with oNIS cells after 2.2 MBq I-131 treatment (*p* < 0.005) (Figure 2C,D). Similarly, both early and late apoptosis increased noticeably after treating either 2.2 or 18.5 MBq I-131 compared to oNIS only treatment (Appendix A). We also confirmed that the amount of DNA fragmentation was obviously increased after I-131 treatment in oNIS+BRD cells, while the amount of DNA fragmentation was consistently low in cells not treated with I-131 (Figure 2E). These results indicate that improved I-131 uptake by oNIS cells reduces cell survival and that the combination of BRG1-BRD with oNIS overexpression may enhance the therapeutic effect of radioiodine by preventing sufficient repair of DNA damage.

### 2.2. Overexpression of oNIS and BRG1-BRD Inhibits Tumor Growth In Vivo through I-131-Enhanced Damage

Thereafter, we considered using BRG1-BRD as a radiosensitizer during radioiodine treatment and established tumor xenograft models using control, oNIS, and oNIS+BRD cells (Appendix A). To visualize tumor growth in the mouse models, we used reporter-expressing FRO cells. A firefly luciferase gene was transduced into each cell line for bioluminescence imaging and tested to confirm that the bioluminescent signal correlated with cell numbers (Appendix A). To determine the most efficient therapeutic dose of I-131, we divided the mice into low- (2.22 MBq) and high- (18.5 MBq) dose groups. In the low-dose treatment group, the mice that received oNIS cells had 22.4% smaller tumor volumes than the control group, and the mice that received oNIS+BRD cells had 74.3% smaller tumor volumes than the oNIS group (*p* < 0.005) (Figure 3A). In the high-dose treatment group, mice that received oNIS cells had tumors that were 68.7% smaller than those in the control group, while mice that had received oNIS+BRD cells had tumors that were 63.1% smaller than those in mice that received oNIS cells (*p* < 0.005) (Figure 3A). The benefit of synergistic oNIS+BRD overexpression in the high-dose group was less apparent than in the low-dose group, because oNIS uptakes I-131 more efficiently than hNIS and 18.5 MBq I-131 already causes a dramatic decrease in tumor volume. However, regarding the low-dose mouse model, a combination treatment with BRD induced a similar level of tumor size reduction with applying only 11.8% of 18.5 MBq, which indicates the possibility of lowering the I-131 dose application for treatment to avoid second malignancy (Figure 3A). In both treatment groups, bioluminescence imaging demonstrated a pattern similar to tumor volumetric measurements (Figure 3A–C). On day 10 after low-dose treatment, the tumors derived from the control, oNIS- and oNIS+BRD-expressing cells grew by 6.66-, 6.09-, and 3.30-fold, respectively (Figure 3B). On day 10 after high-dose treatment, the sizes of the control, oNIS-, and oNIS+BRD-expressing tumors had changed by 2.82-, 0.76-, and 0.19-fold, respectively (Figure 3C). Although no significant differences were observed in body weight in the low-dose group (Appendix A), slight weight loss was observed in the high-dose group (Appendix A) compared with the mice administered with a low dose, which reflected the increased level of overall damage in the mice. Presumably, the increased iodine uptake allowed by oNIS induced more efficient DNA damage, which resulted in the inhibition of tumor growth. These results indicate that BRG1-BRD overexpression and oNIS can induce a synergetic effect for radioiodine treatment in vivo.

### 2.3. Immunostained Tumor Sections Show an Increase in DNA DSBs and Fewer Proliferating Cells after I-131 Treatment in the Presence of oNIS+BRD

In the mouse xenograft model, the size of the extracted tumors decreased significantly after oNIS overexpression followed by BRG1-BRD overexpression (Figure 3D). Histological analysis of tumor sections showed that DNA DSBs, as represented by the γ-H2AX marker, increased 1.4-fold (2.22 MBq) and 1.3-fold (18.5 MBq) in the oNIS+BRD group and the oNIS alone group, respectively, after I-131 treatment (*p* < 0.005) (Figure 3E). Expression of the proliferation marker ki67 was assessed to measure the growth fraction of cells from a xenograft model after radioactive iodine treatment, and its expression was significantly decreased in cells that overexpressed oNIS+BRD (*p* < 0.005) compared with cells that overexpressed oNIS alone in both the low-dose and high-dose I-131 treatment groups, which supports the results observed in Figure 2A,B (Figure 3D,E).

A high dose of I-131 (18.5 MBq) led to more DNA DSBs and higher expression of proliferation markers compared with a low dose of I-131 (Figure 3D,E). The increased γ-H2AX and decreased ki67 indicate that the therapeutic effect of BRG1-BRD overexpression can be maximized with oNIS when 18.5 MBq of I-131 is applied. Representative histology images of tumors stained for myc-tag, γ-H2AX, ki67, and H&E as well as tumor size (Figure 3D) show that BRG1-BRD overexpression in oNIS-expressing cells suppresses cancer cell proliferation and increases DNA DSBs, which enhances the effect of I-131 radiotherapy.

### 2.4. oNIS and BRG1-BRD Overexpression Induces Citrullination while maintaining the Total Citrulline Level and Increasing p21 and NPM1 Expression after I-131 Treatment

We found that 18.5 MBq I-131 induced morphological changes in oNIS+BRD cells, which formed a bridge connecting two adjacent cells (Figure 4A). We assumed that this observation could be extracellular trap (ET)-like structures (or neutrophil extracellular traps (NET) [31] in particular), which are released during NETosis. NETosis is a unique form of cell death caused by histone deimination and the release of decondensed chromatin [32,33]. Histone modification by citrullination is essential for NETosis, which is known to inhibit peptidyl-arginine deiminase 4 (PAD4) enzyme expression [34,35]. Hence, we investigated citrulline expression in the bridge between cells to determine if ET formation was induced by oNIS+BRD and confirmed citrulline expression in the bridge using immunocytochemistry (Figure 4B). This citrulline expression was associated with high-dose radioiodine treatment rather than low-dose treatment (Appendix A). Furthermore, PAD4 enzyme expression was also confirmed (Figure 4C), as PAD4 deficiency is known to impair NETosis by inhibiting the enzymatic conversion of arginine residues that aid in chromatin decondensation [36]. Although we showed that citrulline was released in the oNIS+BRD group (Figure 4B,C), the total amount of citrulline and PAD4 was unchanged by either radioiodine treatment and/or oNIS+BRD gene integration (Figure 4C). However, small fragments of PAD4 (<50 kDa, dotted box in Figure 4C) were not produced after oNIS+BRD overexpression, which indicated that the treatment prevented degradation of the original protein and subsequent production of smaller fragments. Moreover, isoform details from The Cancer Genome Atlas (TCGA) show that thyroid cancer exhibits relatively strong expression of PAD4 isoforms 1 and 5 compared with other cancer types, while none of the other isotypes were overexpressed in thyroid cancer (Appendix A). This result is consistent with a recent report [37], which indicated that PAD4 degradation through the ubiquitin–proteasome pathway inhibits NETosis. Altogether, these findings suggest that oNIS+BRD overexpression induces citrulline crosstalk between cells and maintains intact PAD4, whose higher activation contributes to NETosis.

Based on the finding that γ-H2AX is required for cell cycle arrest via the p53/p21 pathway [38], we expected that oNIS+BRD would also regulate the cell cycle. We evaluated the expression levels of p53/p21 and their phosphorylated forms. The p53 expression level was significantly increased in both oNIS and oNIS+BRD cells after iodine treatment. Consistent with these results, p21, a major target of p53, was also increased in both oNIS and oNIS+BRD cells after I-131 treatment. P21 expression in oNIS+BRD cells was higher than that in oNIS cells (Figure 4D). However, no changes in 7hosphor-p53 (Ser15) and 7hosphor-p21 (Thr145) were observed even after I-131 treatment (Appendix A).

To further explore this mechanism, we focused on the observation that p53 transactivates PAD4 through an intronic p53-binding site. Ectopic expression of p53 or PAD4 was reported to induce protein citrullination and finally lead to the citrullination of the histone chaperon protein NPM1, which results in translocation of NPM1 from the nucleolus to the nucleoplasm [39,40]. Therefore, we observed NPM1 expression in addition to p53, p21, citrulline, and PAD4 expression in oNIS and oNIS+BRD cells with/without I-131 treatment. Consequently, we established that NPM1 expression was significantly increased in oNIS+BRD cells after I-131 treatment (18.5 MBq) (Figure 4D). To determine the function of these differentially expressed proteins, they were subjected to STRING gene ontology [41] analysis and were categorized according to their biological processes (Figure 4E). This analysis showed that p53 (TP53), p21 (CDKN1A), PAD4 (PADI4), and NPM1 are interacting partners during chromatin assembly, response to radiation, mRNA transcription, and response to stress. These results suggested a hypothesis related to the oNIS+BRD therapeutic pathway, as depicted in Figure 4F, and also suggested the following results from PAD4 degradation, induced citrullination including cell morphology, and STRING gene ontology enrichment analysis: (1) degradation of PAD4 is inhibited by oNIS+BRD treatment and induces citrullination; (2) increased NPM1 modulates the stress response and growth suppression by binding and stabilizing p53 in the nucleoplasm, which promotes growth arrest; and (3) p21 is induced by p53 following DNA damage.

## 3. Discussion

Authors should discuss the results and how they can be interpreted from the perspective of previous studies and of the working hypotheses. The findings and their implications should be discussed in the broadest context possible. Future research directions may also be highlighted.

Radioactive iodine therapy using I-131 has been the preferred treatment for thyroid carcinoma and hyperthyroidism since the 1940s [42,43]. Over the past two decades, significant progress has been made in cloning of the human NIS gene in preclinical and translational research [6,7], and these enhanced gene therapy strategies have been studied in both thyroid and nonthyroid tumors [9,10,11,12]. The SLC5A5 gene, which encodes the NIS protein, exhibits no significant patterns across different tissue types (normal vs. tumor) or cancer stages, but SLC5A5 is highly expressed in three specific cancer types: stomach adenocarcinoma (STAD), testicular germ cell tumor (TGCT), and thyroid carcinoma (THCA) (Appendix A). In cancers in general, survival is significantly improved when SLC5A5 is highly expressed compared with when its expression is low (*p* = 0.0019) (Appendix A). NIS is not only a powerful therapeutic gene, but it is also an important imaging reporter for molecular therapies that can lead to more accurate therapeutic applications [8,10]. However, some patients whose tumors already exhibit strong NIS expression may not require additional NIS expression, and for those who are able to take up iodine efficiently with their own functional NIS, the application of BRG1-BRD overexpression alone will be advantageous. As shown in Appendix A, most cancer types lack sufficient NIS expression that could bring benefits from oNIS+BRD therapy. Therefore, this strategy of BRD or oNIS+BRD overexpression could offer more therapeutic options based on patients’ endogenous NIS expression in their tumors, which can also be monitored by the NIS reporter gene.

Although I-131 therapy has led to tremendous benefits in patients with thyroid diseases including cancer, I-131 therapy still has limitations, including the development of autoimmune diseases as a result of the ectopic increase in helper T cells [44,45]. To reduce the risks of I-131, minimal doses of radioiodine are used, and various studies have been performed to optimize the hNIS gene to increase iodine uptake [14], to add radiosensitizers [46], and to inhibit telomerase reverse transcriptase expression to improve the antitumor effects [47]; other innovative targeted radionuclide therapies have also been reported [48,49,50,51,52]. New strategies to prevent unexpected secondary diseases resulting from radioactive iodine accumulation are needed. Previously, we reported that radioiodine uptake by oNIS-expressing tumors was 2.3-fold greater than that in human NIS-expressing tumors [14]. We also reported that BRG1-BRD, which inhibits binding of the SWI/SNF complex to histones and acts as a radiosensitizer upon overexpression, delayed tumor growth up to 2.7-fold in vivo after high-energy X-ray treatment (9 Gy, day 28) [30]. Based on our previous research, we developed a method to merge these two effective therapeutic genes, oNIS and BRG1-BRD, to improve the antitumor effects of radioactive iodine treatment [14,30].

I-131 therapy has been generally applied in the range of 3.7–7.4 MBq per gram of thyroid [53] or in the range of 0.925–7.4 GBq [54] in the clinic. When a tumor has received a cumulative dose greater than 22.2 GBq of radioiodine without signs of remission, this is defined as the radioiodine-refractory phase, which can lead to genetic mutations and radiation resistance [55]. Thus, it is important to limit the dose of radioactive iodine to avoid unnecessary exposure while still achieving the same therapeutic effect. For example, it has been reported that cotreatment with thyrotropin alfa can reduce the dose of I-131 to 1.1 GBq, which is similar to the effect seen with 3.7 GBq I-131 alone [16].

In this study, we demonstrated the therapeutic efficacy of two doses (2.22 and 18.5 MBq) in both cancer cells and mouse tumor models with oNIS or oNIS+BRD overexpression. Our strategy was to use the oNIS or oNIS+BRD genes as therapeutic genes in tumor cells to enhance the therapeutic effect of I-131 through an increase in iodine absorption and an enhancement in radiation sensitivity. By establishing oNIS- and oNIS+BRD-expressing cell lines, we constructed cell line models in which these therapeutic genes (oNIS and BRD) were introduced and it was confirmed that the morphology and division rate of these cells are the same as those of the parental cell line (FRO) (Figure 1). Our previous study focused on animal models and evaluated iodine uptake levels and the therapeutic efficacy of oNIS [14]; we also validated cell survival and performed a γ-H2AX foci analysis after I-131 therapy. Since γ-H2AX plays an important role in initiating DNA DSB repair [56] by mediating cluster formation of specific proteins at the DSB, as well as by inducing H3 acetylation at the binding site of BRG1-BRD [29], DNA damage repair can be measured by γ-H2AX foci analysis. Our data showed that oNIS overexpression, which increased iodine uptake through codon optimization, was associated with increased γ-H2AX foci and DNA fragmentation, which led to lower cell survival after I-131 treatment (Figure 2). In tumor models, the number of γ-H2AX foci observed after quantification increased with BRD expression as well as with the therapeutic dose of radioiodine therapy in vivo (Figure 3). These results demonstrated that BRD expression increases DNA DSBs in response to oNIS-specific radioiodine therapy and enhanced the radiosensitizing effect. Therefore, if oNIS+BRD expression could be selectively regulated using an oncolytic virus for gene delivery, such as talimogene laherparepvec (T-Vec), a second-generation oncolytic herpes simplex virus type 1 (HSV-1) armed with GM-CSF [57,58], normal tissue with high endogenous NIS expression would be spared from unnecessary damage associated with increased iodine uptake, and damage induced by oNIS+BRD overexpression would be specific to cancer cells.

Since the BRD of histone acetyltransferase (HAT) p300/CREB binding protein-associated factor (PCAF) recognizes acetyl-Lys moieties [59], the interaction between BRG1-BRD and acetylated H3 is necessary for DNA DSB repair [29]. This histone acetylation at DSBs is associated with citrullination [60] and the recruitment of chromatin-binding proteins such as histone chaperones. Citrullination, which is the conversion of arginine in a protein into citrulline by deimination, plays a role in various types of cell deaths including apoptosis, autophagy, and NETosis [61]. Recently, the switch between apoptosis and NETosis [62,63] and the simultaneous induction of apoptosis and NETosis, termed apoNETosis [64,65], have both been reported. Citrullination of histones, which is mediated by PAD4 expression, plays a significant role in post-translational modifications and specifically triggers the formation of a highly condensed NET-like chromatin structure [34]. Extrusion of NET-like structures from cells is a signature of NETosis-like cell death initiation. In response to high-dose (18.5 MBq) radiation, we observed an NET-like structure (Figure 4A,B) and citrulline expression (Figure 4B) in oNIS+BRD cells. PAD4, an enzyme that converts arginine or monomethyl-arginine to citrulline in histones, plays a crucial role in NET formation [66], but the total levels of citrulline and PAD4 did not change regardless of exogenous gene expression and radioiodine treatment (Figure 4C). In addition, PAD4 serves as a p53 corepressor to regulate histone arginine methylation at the p53-target gene p21/WAF1/CIP1 promoter [67]. Therefore, we investigated p53 and p21 expression after radioiodine treatment (Figure 4D). At the genomic level, p53, p21, and NPM1 are highly expressed in most tumors including thyroid cancer, whereas PAD4 levels are low, except in acute myeloid leukemia and thymoma (Appendix A). As a multifunctional histone protein and a substrate of PAD4 for citrullination [68], NPM1 accumulates rapidly at the site of DNA DSBs and has a crucial role in DDR [69,70,71,72]. It is reported that knocking down NPM1 reduced tumor cell survival significantly after radiation [73]. Consistent with this result, Traver et al. showed that YTR107, a small molecule suppressing pentamer formation of NPM1, inhibits RAD51 formation followed by a sensitized radiotherapeutic effect [72]. NPM1 is also acetylated by p300 acetyltransferase, which enhances its affinity to acetylated histones; this is essential for transcriptional activation by chromatin decondensation [74,75]. As acetylation of Lys 14 in H3 is associated with NPM1 [75] and increased NPM1 in oNIS or oNIS+BRD cells, it might be related to the chromatin remodeling process (Figure 4D). Our results also showed that I-131 treatment increased the levels of total p53 and p21 in oNIS- and oNIS+BRD-expressing cells after radioiodine treatment. We could then speculate how BRD overexpression is involved in increasing p53, p21, and NPM1 expression when used as a radiosensitizer. Our results are supported by those of Tanikawa et al. [39], who demonstrated that the p53-PAD4 pathway regulates the shuffling of NPM1 via citrullination and plays critical roles in p53-dependent pathways. Based on previous studies, we propose the following radiosensitizing mechanism by oNIS+BRD gene overexpression. Ectopic expression of oNIS with BRD increases NPM1 expression, which is citrullinated by PAD4 in nucleoli [40], which results in p53 stabilization. Consequently, the p53 complex translocates to the nucleoplasm and induces the p53/p21 tumor suppressor pathway. Even though there has had considerable progress in gene therapy during in recent decades, it is still challenging to deliver therapeutic nucleic acids into target cells in order to achieve a beneficial therapeutic effect. In the thyroid gland, the NIS gene is expressed at a high level. Thus, labelled of therapeutic nucleic acids with iodine will be delivered selectively to the thyroid gland though iodine uptake.

Collectively, we evaluated the enhancement of the therapeutic effect induced by oNIS and BRD gene overexpression after I-131 treatment and partially elucidated the mechanism. Since tumor-targeting viral vectors, including oncolytic viral vectors, are efficiently used to deliver genetic materials to target cells [76], oNIS+BRD overexpression in target tumor cells has therapeutic potential for the treatment of various cancers, including thyroid cancer, with heterogenous hNIS expression. Our findings provide insight into improved I-131 therapy by combining efficient iodine uptake by oNIS with hampered DNA damage repair by BRG1-BRD.

## 4. Materials and Methods

### 4.1. Cell Lines

FRO, a human anaplastic thyroid cancer cell line that lacks NIS expression, was used for this experiment. For tumor visualization, FRO cells expressing firefly luciferase were used. FRO-Luc (control), FRO-Luc-expressing optimized-sodium iodide symporter (oNIS), and FRO-Luc-expressing oNIS and BRG1-bromodomain (oNIS+BRD) cells were maintained in RPMI-1640 medium (WelGene Inc., Seoul, Republic of Korea) supplemented with 5% fetal bovine serum (FBS) and 1% penicillin/streptomycin at 37 °C and 5% CO_2_.

### 4.2. Retroviral Transduction

oNIS and oNIS+BRD cells were established according to previous methods [30]. Retroviruses were generated using a modification of the Retro-X universal packaging system (BD Biosciences Clontech, Palo Alto, CA, USA); 293FT packaging cells were cotransfected with 5 μg of pMSCV/opt-NIS vector, 5 μg of pcDNA3 gag-pol vector, and 5 μg of envelope vector using Lipofectamine^®^ 2000 (Invitrogen, Carlsbad, CA, USA). Control cells expressing oNIS were maintained after puromycin selection, and monoclonal cells with a similar number of oNIS transcripts were isolated for further analysis. For BRD overexpression, BRD containing myc-tag, to detect expression of recombinant protein, was transfected into oNIS cells for 2 days, after which oNIS+BRD cells underwent monoclonal selection. The transfection efficiency of these cells was evaluated to measure photon flux signals using an in vivo imaging system (IVIS) 100 and real-time RT–qPCR.

### 4.3. Quantitative Real-Time Reverse Transcription–Polymerase Chain Reaction (Real-Time RT–qPCR)

Total RNA was isolated from cultured cells using TRIzol (Qiagen, Valencia, CA, USA) according to the manufacturer’s protocol. cDNA was synthesized from total RNA using gene-specific primers according to the TaqMan RT master mix protocol (PE Applied Biosystems, Foster City, CA, USA). Real-time RT–qPCR was performed using an Applied Biosystems 7500 Sequence Detection system. Reactions were incubated in a 96-well optical plate at 50 °C for 2 min and 95 °C for 10 min, followed by 40 cycles at 95 °C for 15 s and 60 °C for 10 min. The mRNA level of myc-tag was determined using the following primers for fluorescence detection: forward 5′-GGTAGATACGGCCGCAGAA-3′ and reverse 5′-CGGCCCCATTCAGATCCT-3′, designed with FAM; 5′-CTTCTGAGATGAGTTTTTG-3′. Glyceraldehyde-3-phosphate dehydrogenase (GAPDH) was used as an internal control.

### 4.4. Iodine Uptake Assay

Radioactive iodine (I-125) and I-131 were purchased from the Korea Atomic Energy Research Institute (KAERI, Daejeon, Republic of Korea) in the form of sodium iodide. To confirm the efficacy of oNIS overexpression in each group of cells, an I-125 uptake assay was performed in vitro. Each group of cells (1 × 10^5^) was plated in triplicate in 24-well plates, washed with warmed Hank’s balanced salt solution (HBSS), and then incubated with 500 μL of warmed HBSS containing 3.7 kBq of I-125 and 10 μM nonradioactive NaI for 30 min at 37 °C. These cells were washed twice with cold HBSS and lysed for 5 min in 1% sodium dodecyl sulfate (SDS). Cell lysates were collected, and radioactivity was measured using a gamma counter (Canberra-Packard, Meriden, CT, USA). Radioactivity was normalized to the total amount of protein in each tube.

### 4.5. Doubling Time

Each group of cells (5 × 10^4^) was plated in triplicate in 6-well plates and harvested at 24 h, 48 h, and 72 h. These cells were stained with trypan blue and counted using a hemocytometer. Doubling times of harvested cells were calculated using an algorithm available in GraphPad Prism (GraphPad Software, La Jolla, CA, USA): TD = t × lg2/(lgNt − lgN0), where N0 is the number of cells inoculated, Nt is the number of cells harvested, and t is the culture time in hr.

### 4.6. Clonogenic Assay

Each group of cells (2 × 10^2^) was plated in triplicate in 6-well plates. Cells were treated with I-131 at various doses (0, 0.55, 1.11, and 2.22 MBq) for 7 h. After 10 days, colonies of irradiated cells were stained with a 0.5 g crystal violet (Sigma Aldrich, St. Louis, MO, USA) solution. The colonies were counted, and the mean value and standard deviation were calculated.

### 4.7. Immunocytochemistry (ICC) for γ-H2AX Foci and Citrulline Staining

Each group of cells (5 × 10^4^) was incubated with 2.22 MBq or 18.5 MBq of I-131 for 7 h and then fixed in methanol at −20 °C for 5 min. After blocking with normal goat serum (Vector Laboratories, Inc., Burlingame, CA, USA) for 30 min, cells were incubated with the following primary antibodies at 4 °C overnight: anti-γ-H2AX antibody (Cell Signaling, Danvers, MA, USA) and anti-histone H3 (citrulline R2 + R8 + R17) (Abcam, Cambridge UK; ab5103). The cells were washed and stained with Alexa Fluor 555 goat anti-rabbit IgG (Invitrogen, Carlsbad, CA, USA) for 1 h at room temperature and then mounted with ProLong Gold antifade reagent with DAPI (Life Technologies, Grand Island, NY, USA). Fluorescence was observed using a confocal microscope (Carl Zeiss, Oberkochen, Germany).

### 4.8. DNA Fragmentation Assay

Each group of cells (5 × 10^6^ cells per group) was incubated with 2.22 MBq of I-131 for 7 h, and DNA digested with proteinase K was extracted with a G-DEX IIc genomic DNA extraction kit (Intron Biotechnology, Seongnam, Republic of Korea) according to the manufacturer’s recommendations. Extracted DNA was run on a 1% agarose gel at 100 V for 30 min in TAE buffer.

### 4.9. Western Blotting

Cells were lysed in RIPA buffer (Sigma-Aldrich Corporation, St. Louis, MO, USA) containing protease inhibitor cocktail (Roche, Cher, Centre, France). The concentration of the extracted protein was analyzed using a BCA assay (Pierce, Rockford, IL, USA). The obtained protein (30 μg) was separated by SDS–PAGE under reducing conditions and transferred onto a PVDF membrane (Millipore, Billerica, MA, USA). After blocking with 5% skim milk or bovine serum albumin (BSA) for 1 h, membranes were incubated with the following antibodies: anti-myc-tag (Cell Signaling, Danvers, MA, USA), anti-hNIS (Koma Biotech, Seoul, Republic of Korea), anti-citrulline (Merck, Darmstadt, Germany), anti-p53 (Calbiochem, Darmstadt, Germany), anti-p-p53 (Cell Signaling, Danvers, MA, USA), anti-NPM1 (Santa Cruz, Dallas, TX, USA), anti-PAD4 (Abcam, Cambridge, UK), anti-p21 (Cell Signaling, Danvers, MA, USA), and anti-p-p21 (Santa Cruz, Dallas, TX, USA). This was followed by incubation with horseradish peroxidase-conjugated anti-mouse (Cell Signaling, Danvers, MA, USA) or anti-rabbit (Cell Signaling, Danvers, MA, USA) secondary antibody for 1 h at room temperature. β-actin (Sigma–Aldrich, St. Louis, MO, USA) was used as a loading control. Protein levels were detected by enhanced chemiluminescence (Roche, Cher, Centre, France) using an LAS-3000 Imaging System (Fujifilm, Cypress, CA, USA).

### 4.10. Tumor Xenografts in Nude Mice

All procedures involving animals were approved by the Institutional Animal Care and Use Committee (IACUC) at Seoul National University Hospital, Republic of Korea, and were consistent with the Guide for the Care and Use of Laboratory Animals [77]. In this study, 5 × 10^5^ cells per group (n = 5) were mixed with 40 μL of Matrigel and were subcutaneously injected into the flank of male BALB/c-nu/nu mice (7 weeks of age).

### 4.11. Therapeutic Effect of I-131 In Vivo

The therapeutic effect on tumor-bearing mice was evaluated by two methods: measurement of the tumor size and bioluminescence. Tumor size was measured weekly using a caliper and was estimated using the formula: tumor volume = length × width2/2, where the length represents the largest tumor diameter, and the width represents the perpendicular tumor diameter. Tumor-bearing mice were intraperitoneally (i.p.) administered either 2.22 or 18.5 MBq I-131. Bioluminescence imaging and tumor measurements were performed at 0, 2, 6, and 10 days.

### 4.12. Bioluminescence Imaging

An IVIS 100 imaging system (Calliper Life Sciences, Hopkinton, MA, USA) with an optical CCD camera mounted in a light-tight specimen chamber was used for bioluminescence image acquisition. D-luciferin potassium salt was diluted to 0.3 mg/mL in PBS before use, and 100 μL of the D-luciferin solution was intraperitoneally injected into the mice. Bioluminescence images were serially acquired every 5 min until the maximal signal was recorded. To quantify emitted light, a region of interest (ROI) was drawn over the tumor region. Total photon flux was expressed as photons per cm^2^ per second per steradian (p/cm^2^/s/sr).

### 4.13. Immunohistochemistry

Tissues were fixed in 10% neutral-buffered formalin, processed, and embedded in paraffin. Slides were deparaffinized in xylene and rehydrated in graded alcohol solutions (from 100% to 70%) and distilled water for 5 min in each solution. Methanol with 0.5% hydrogen peroxide was used to block endogenous peroxidase activity. Slides were then immersed in 10 mM sodium citrate (pH 6.0) and boiled for heat-induced epitope retrieval. After the slides were allowed to cool to room temperature, they were rinsed in 0.5% Triton X-100 in PBS. Non-specific proteins were blocked by incubating the sections in normal goat serum in PBS (1:30) for 30 min. The slides were incubated with the following primary antibodies at 4 °C overnight: anti-myc-tag (Cell Signaling, Danvers, MA, USA), anti-ki67 (Abcam, Cambridge, UK), and anti-γ-H2AX (Cell Signaling, Danvers, MA, USA). The sections were incubated with biotinylated secondary antibodies for one hour at room temperature, and ABC complex binding was performed before DAB and hematoxylin staining. For quantification, stained tissues were imaged by TissueFAXS, and the results were calculated using TissueQuest software (TissueGnostics, Vienna, Austria).

### 4.14. Statistical Analysis

The significance of the differences between measurements was evaluated by the Mann–Whitney U test (*p* < 0.05), which was used for both in vitro and in vivo experiments. All experiments were performed in triplicate. The gene expression profiles were analyzed using the Gene Expression Profiling Interactive Analysis (GEPIA) database.

## 5. Conclusions

In this study, we evaluated the effect of enhanced radioiodine therapy by overexpressing both the oNIS and BRG1-BRD genes. oNIS overexpression efficiently increased radioiodine uptake levels in cancer cells and reduced the required I-131 dose for effective therapy. I-131 treatment in tumor cells induced increased DNA damage when coupled with BRG1-BRD overexpression, which interfered with the function of the BRG1 chromatin remodeling enzyme. Specifically, we found that BRG1-BRD overexpression increased NPM1 expression when cells were treated with I-131, thereby stabilizing the p53-dependent pathway. The enhancement in radiation sensitivity by oNIS+BRD gene therapy may be beneficial to those who cannot receive a high dose of radiation due to potential risks, as a lower dose of radioiodine can be applied to achieve the equivalent therapeutic effect.

## Figures and Tables

**Figure 1 ijms-24-02761-f001:**
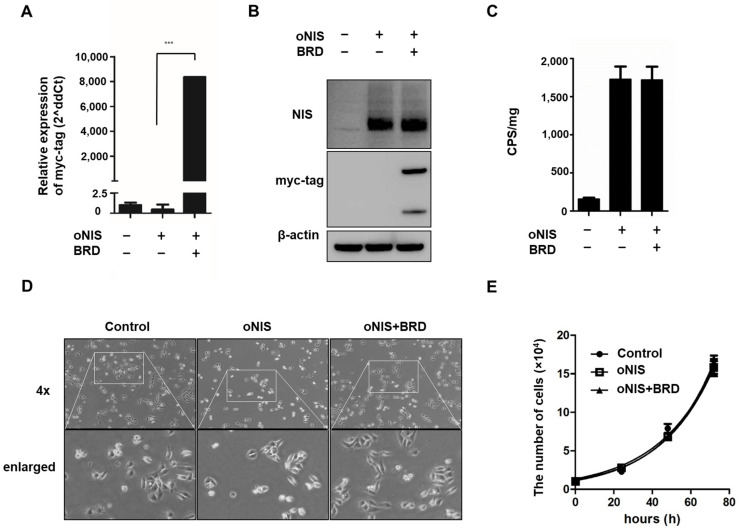
Establishment of oNIS and oNIS+BRD cells. (**A**) Transcriptional levels of BRG1-BRD were quantified by real-time RT–qPCR. The qPCR data were normalized by GAPDH. Data are presented as the means ± S.D (n = 4). (**B**) oNIS or oNIS+BRD overexpression was evaluated by Western blot. (**C**) Transfection efficiency was evaluated by an I-125 uptake assay. CPS is counts per second. (**D**) Microscopy images of nontransfected control, oNIS-, and oNIS+BRD-expressing cells. The morphology did not change after viral infection. A 4× scanning objective lens was used which gives a total 40× magnification. (**E**) Measurement of the cell proliferation rate among the control, oNIS-, and oNIS+BRD-expressing cells. Values are means ± S.E.M. *p* < 0.001 (***).

**Figure 2 ijms-24-02761-f002:**
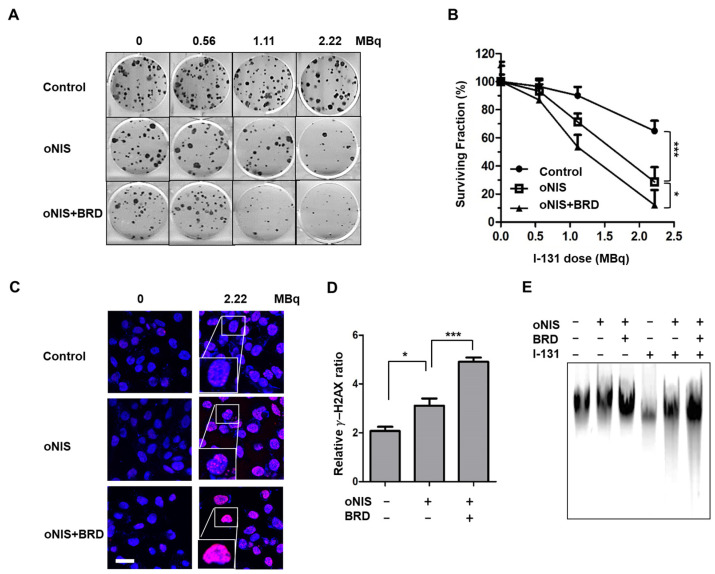
Therapeutic effect according to BRD overexpression in response to I-131 treatment. (**A**) Clonogenic survival assay in control, oNIS-, and oNIS+BRD-expressing cells. Cells were treated with 0, 0.56, 1.11, or 2.22 MBq of liquid I-131 for 7 h, washed, and incubated in 6-well plates for 10 days. (**B**) Quantification of the clonogenic survival assays in control, oNIS, and oNIS+BRD cells. (**C**) Representative fluorescence images of γ-H2AX foci. Cells were immunostained with anti-γ-H2AX (red) and DAPI (blue) after treatment with liquid I-131 (2.22 MBq) after a 7 h incubation. Scale bar = 30 μm. (**D**) Quantification of the γ-H2AX foci obtained from fluorescence confocal images. (**E**) DNA fragmentation assay with or without 2.22 MBq I-131 treatment. DNA isolated from cells was run on a 1% agarose gel at a current of 100 V for 30 min in TAE buffer. Values are means ± S.E.M. *p* < 0.001 (***), *p* < 0.05 (*).

**Figure 3 ijms-24-02761-f003:**
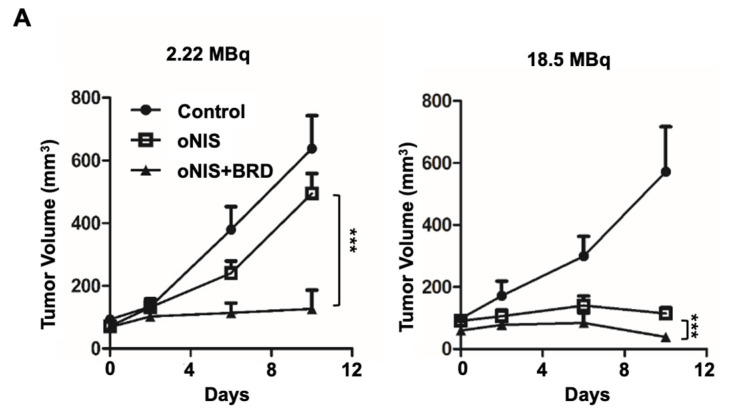
Radiation sensitization by BRD after low-dose (2.22 MBq) or high-dose (18.5 MBq) I-131 treatment. (**A**) Tumor volumes were measured with a caliper up to 10 days after treatment with 2.22 MBq or 18.5 MBq I-131. (**B**,**C**) A region of interest (ROI) was measured to quantify the luminescent signal from the tumors. The bioluminescence signal of the IVIS 100 correlated with the cell number after I-131 treatment. The scale bar represents the peak signal in photons/s/cm^2^/sr. (**D**) Tumor sections (4 μm in thickness) were stained with antibodies against myc-tag, ki67, and γ-H2AX. Scale bar for IHC = 50 μm. Scale bar for tissue = 2 mm. (**E**) Quantification of γ-H2AX or ki67 intensity in tumor tissues from D using TissueFAXS software. Values are means ± S.E.M. *p* < 0.001 (***), *p* < 0.05 (*).

**Figure 4 ijms-24-02761-f004:**
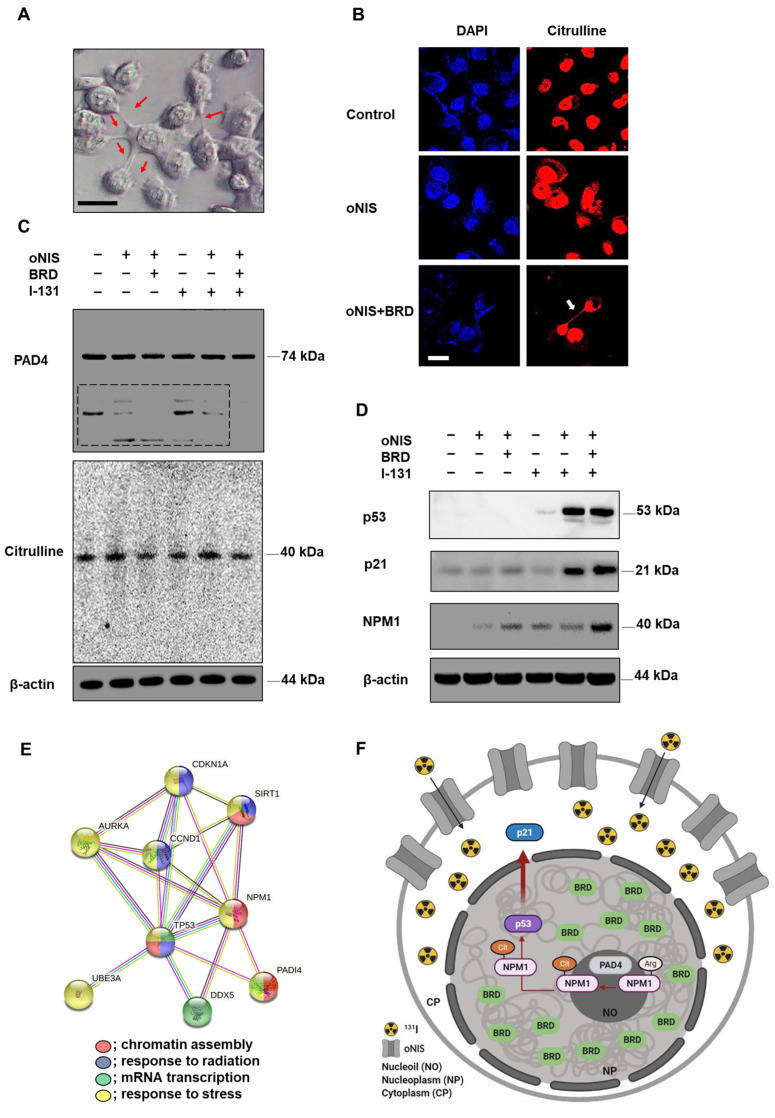
Observation of extracellular trap-like structures and post-translational modifications of histones dependent on NPM1 expression after treatment with 18.5 MBq I-131. (**A**) Representative DIC image (100× objective) of oNIS+BRD cells after I-131 treatment. Scale bar = 30 μm. (**B**) Immunocytochemistry using citrulline antibodies and DAPI. Decondensed chromatin structure was stained using a citrulline antibody (100× objective). Scale bar = 30 μm. (**C**) Protein expression of citrulline and PAD4 was evaluated by Western blot. (**D**) Protein expression of NPM1 and tumor suppressors, such as p53 and p21, was evaluated by Western blot. (**E**) STRING Gene Ontology enrichment analysis of biological processes. There is an intricate protein–protein interaction network between PAD4 and NPM1. (Red: chromatin assembly, blue: response to radiation, green: mRNA transcription, yellow: response to stress.) (**F**) A diagram depicting the radiosensitizing mechanism. oNIS with BRD overexpression increases NPM1 expression, which is citrullinated by PAD4 in nucleoli and results in p53 stabilization. Consequently, the p53 complex translocates to the nucleoplasm, which further induces the p53/p21 tumor suppressor pathway.

## Data Availability

Not applicable.

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
