# Peer review of "Overexpression of Both Human Sodium Iodide Symporter (NIS) and BRG1-Bromodomain Synergistically Enhances Radioiodine Sensitivity by Stabilizing p53 through NPM1 Expression"

_ijms, 2023, doi:10.3390/ijms24032761_

Round 1
Reviewer 1 Report
Summary: In this study, authors showed that oNIS+BRD plays important role in radioiodine treatment. The synergistic effect of both BRG1-BRD and oNIS gene overexpression resulted in more DNA double-strand breaks and led to reduced cell proliferation in radiation treatment. This study observation will add more information for improving the radiation therapy.
Comments:
1.What is the effect of oNIS+BRD overexpression on cell cycle or apoptosis after I-131 treatment?
2.Tumor volume is significantly reduced in oNIS +BRD animals. But 18.5 MBq I-131
treatment there is no significant difference between oNIS overexpression vs oNIS+BRD. Author should explain the observation.
3. Apoptosis or cycle markers needs add in IHC for more conformation of the I-131 treatment effect.
4. Authors should explore the effect of I-131 treatment in absence of NPM1 in oNIS +BRD overexpression cells.
5. How oNIS +BRD overexpression helps in translocation of NPM1? Need some experimental evidence.
Author Response
Response to reviewers’ comments
We wish to thank both reviewers for their careful consideration of our manuscript and their constructive comments. We have taken this opportunity, prompted by the reviewers’ comments to perform the additional experiments and update. These new experiments and the other changes detailed below we believe have addressed the reviewers’ concerns and collectively these changes strengthen the manuscript.
Reviewer #1
Comments and Suggestions for Authors
Summary: In this study, authors showed that oNIS+BRD plays important role in radioiodine treatment. The synergistic effect of both BRG1-BRD and oNIS gene overexpression resulted in more DNA double-strand breaks and led to reduced cell proliferation in radiation treatment. This study observation will add more information for improving the radiation therapy.
We thank the reviewer for their positive comments.
Comments:
1.What is the effect of oNIS+BRD overexpression on cell cycle or apoptosis after I-131 treatment?
We accept reviewer’s point because the loss of apoptotic control allows cancer cells to survive longer. Therefore, we treated either 2.2 or 18.5 MBq of I-131 on FRO wt, oNIS, or oNIS+BRD cells. We were able to detect increased apoptosis level in oNIS+BRD compared to oNIS cells. In the manuscript, the apoptosis assay (Annexin-V/PI) results was in supplementary figure 1 and a sentence was also added in line 96-98, which now reads.
‘Similarly, both early and late apoptosis increased noticeably after treating either 2.2 or 18.5 MBq I-131 compared to oNIS only treatment (Figure S1).’
2.Tumor volume is significantly reduced in oNIS +BRD animals. But 18.5 MBq I-131
treatment there is no significant difference between oNIS overexpression vs oNIS+BRD. Author should explain the observation.
We agree with the reviewer and are happy to clarify this point further in the text. Although the reduced level of tumor size was not huge in 18.5 MBq I-131 treatment mouse model, there was a significant p-value (*** <0.001) in reduced tumor size. The main reason for having different tumour-reduced level in different dose of I-131 treatment (oNIS+BRD vs. oNIS ) is that the imbalance of damages caused by oNIS and BRD, respectively. Codon optimized-NIS, oNIS, uptake I-131 around 4-5.4 times more efficiently than human NIS does, and this higher uptake of I-131 boosted the effect of I-131 therapy. This might make us to think BRD overexpression effect is too minor to apply, however, BRD overexpression hampered the size of tumor in a low-dose of I-131 (2.2 MBq) therapeutic mouse model. Currently, I-131 treatment often causes second malignancy and severe side effects. The purpose of combination therapy is lowering I-131 dose to avoid unnecessary damages during treatment plan. Therefore, the following has been modified and added to the line 119-125.
‘The benefit of synergistic oNIS+BRD overexpression in the high-dose group was less apparent than in the low-dose group, because oNIS uptake I-131 more efficiently than hNIS and 18.5 MBq I-131 already causes a dramatic decrease in tumor volume. However, regarding low-dose mouse model, combination treatment with BRD induced similar level of tumor size reduction with applying only 11.8% of 18.5 MBq, which suggests the possibility of lowering the I-131 dose application for treatment to avoid second malignancy.’
- Apoptosis or cycle markers needs add in IHC for more conformation of the I-131 treatment effect.
We thank the reviewer for highlighting this, however, we were not able to manage extra experiments. Alternatively, we tested a long-term I-131 incubation (5 days) with cells, which are resembled the mouse model timeline (Supplementary Figure 1 C-D). As mentioned in the first comment’s answer, apoptosis level noticeably increased in oNIS+BRD compare to oNIS.
In terms of cell cycle markers, we have shown Ki67 expressions through IHC in Figure 3D-E. Ki67 is recently been reported as an indirect cell cycle marker as it accumulates only during S, G2, and M phases and it degrades continuously in G1 and G0 phases, regardless of the cause of entry into G0/quiescence (Miller et al, cell reports, 2018). We hope Annexin-V/PI data with long-term incubation and the reference provided the confirmation of the I-131 treatment effect.
- Authors should explore the effect of I-131 treatment in absence of NPM1 in oNIS +BRD overexpression cells.
I-131 emits radiation in the form of gamma rays and beta particles. In absence of NPM1, significantly reduced tumor cell survival after radiation treatment (Cs-137, emits both gamma rays and beta particles) was reported (Figure 2 in Wiesmann et al, Trans Onc. 2019). Consistent with this result, Traver et al. showed YTR107, a small molecule suppressing pentamer formation of NPM1, inhibits RAD51 formation followed by a sensitized radiotherapeutic effect (Traver et al, Cancer letter, 2021). Albeit we were not able to approach this with our own experimental results, we could expect oNIS+BRD overexpression cells in absence of NPM1 would result in a higher reduction of cell survival and contributes to permanent damages to DNA repair mechanism based on works of literature. Therefore, the following has been modified and added to the line 364-367.
‘It is reported that knocking down NPM1 reduced tumor cell survival significantly after radiation [73]. Consistent with this result, Traver et al. showed YTR107, a small molecule suppressing pentamer formation of NPM1, inhibits RAD51 formation followed by a sensitized radiotherapeutic effect [72].’
- How oNIS +BRD overexpression helps in translocation of NPM1? Need some experimental evidence.
Cellular stress such as radiation or drugs interfering with the synthesis of rRNA causes the loss of nucleolar integrity, which leads to NPM1 localized in the nucleolus to inhibit HMDM2, stabilize p53, and arrest the cell cycle (Rau&Brown et al, Hematol Oncol. 2009, Dermani et al, J Cell Physiol. 2021). Additionally, PAD4 induces nucleoli/nucleoplasm translocation of NPM1 (Tanikawa et al. Cancer Res, 2009), therefore, inhibited degradation of PAD4 in oNIS+BRD cells (Fig. 4C) would be indirect experimental evidence of NPM1 translocation. oNIS contributes to higher uptake of I-131, and BRD contributes to inefficient DNA damage repairs to induce cell death. Therefore, we could expect NPM1 accumulation in the nucleolus as a result of high cellular damages received.

Reviewer 2 Report
Authors showed that co-expression of oNIS and BRG1-BRD enhanced the radiation sensitivity in human thyroid cancer cells via PAD4-mediated MPM1 citrullination and subsequent activation of p53/p21-dependent pathways. Although the achievements shown in the current manuscript would potentially contribute to the development of a new therapy for thyroid cancers through a combination of radiotherapy and gene therapy, there are several concerns that must be addressed before publication in IJMS.
Major concerns:
1) Direct evidence to show radiation-indued citrullination of MPM1 in oNIS+BRD co-expressed cells is required. Please add the corresponding data in Figure 4D.
2) Direct evidence to show the prevention of PAD4 degradation and an enhancement in PAD4 activity in irradiated oNIS+BRD co-rexpressed cells is required. Please add the corresponding data in Figure 4C.
Minor concerns:
1) For the development of new therapeutics via combination of radiotherapy and gene therapy, cancer-specific expression of oNIS and BRD is required. Please add descriptions regarding the solution of this issue in Discussion.
Author Response
Response to reviewers’ comments
We wish to thank both reviewers for their careful consideration of our manuscript and their constructive comments. We have taken this opportunity, prompted by the reviewers’ comments to perform the additional experiments and update. These new experiments and the other changes detailed below we believe have addressed the reviewers’ concerns and collectively these changes strengthen the manuscript.
Reviewer #2
Comments and Suggestions for Authors
Authors showed that co-expression of oNIS and BRG1-BRD enhanced the radiation sensitivity in human thyroid cancer cells via PAD4-mediated MPM1 citrullination and subsequent activation of p53/p21-dependent pathways. Although the achievements shown in the current manuscript would potentially contribute to the development of a new therapy for thyroid cancers through a combination of radiotherapy and gene therapy, there are several concerns that must be addressed before publication in IJMS.
We thank the reviewer for their comments.
Major concerns:
1) Direct evidence to show radiation-indued citrullination of MPM1 in oNIS+BRD co-expressed cells is required. Please add the corresponding data in Figure 4D.
We agree with reviewer’s point, however, we were not able to manage this extra experiments and there is a supporting literature to elucidate this. According to Tanikawa et al, activated p53 followed by PAD4 activation citrullinate NPM1 under physiologic condition (Cancer Res. 2009). Figure 4 in Tanikawa et al shows the direct binding of PAD4 and NPM1 which indicate that NPM1 is a substrate of PAD4. Increased citrullination of NPM1 after inducing DNA damage by treating doxorubicin was also shown in figure 5 in Tanikawa et al. It is broadly well known that p53 is activated by a variety of cellular stresses, including radiation. Thus, activated p53 by radiation and gene therapy (oNIS+BRD) would inhibits PAD4 enzyme degradation (Fig. 4C in our manuscript), and this actively remained PAD4 will increase citrullination of NPM1.
2) Direct evidence to show the prevention of PAD4 degradation and an enhancement in PAD4 activity in irradiated oNIS+BRD co-rexpressed cells is required. Please add the corresponding data in Figure 4C.
We thank reviewer for their helpful suggestion. Unfortunately, we were not able to generate extra figure regarding this. However, we found a few supporting rationales from literatures. Enhanced PAD4 activity (when PAD4 is activated) can drive the formation of NETs (Liu et al, J Thromb Haemost, 2021; Rohrbach et al, Front. Immunol. 2012; Hemmers et al, PLoS One. 2011; Li et al, J Exp Med. 2010), and figure 4A-B depicted NET formation in oNIS+BRD coexpressed cells after radiation. Ou et al. recently demonstrated that E3 ligase promotes the degradation of PAD4 through ubiquitin proteasome pathway which suppresses NETosis in the end (Ou et al. Nature Comm. 2021). Therefore, we presume that it is in a flow saying that prevention of PAD4 degradation (Figure 4C) which is promoted by radiation and oNIS+BRD maintained PAD4 enzymatic activity. And this active PAD4 drove NETosis (Figure 4A-B).
Minor concerns:
1) For the development of new therapeutics via combination of radiotherapy and gene therapy, cancer-specific expression of oNIS and BRD is required. Please add descriptions regarding the solution of this issue in Discussion.
I understood this comment as an application/drug delivery question, please correct me if I misunderstood the comment.
Even though there has had considerable progress in gene therapy during last decades, it is still challenging to deliver therapeutic nucleic acids into target cells in order to achieve a beneficial therapeutic effect. A specific carrier or genetic vehicle, such as synthetic materials (polymers, lipid, lipid nanoparticles) that encapsulate therapeutic nucleic acids, improved delivery to desired cells or tissues so far. In our study, we are focused on thyroid cancer, thus iodine-labelling would deliver our therapeutic nucleic acids through sodium iodide symporter (NIS). As shown in figure S6A, thyroid has outstanding NIS expression compared to any other organs. Here, a few sentences were added in line 382-386, which now reads:
‘Even though there has had considerable progress in gene therapy during last decades, it is still challenging to deliver therapeutic nucleic acids into target cells in order to achieve a beneficial therapeutic effect. In thyroid gland, NIS gene is expressed at a high level. Thus, labelled of therapeutic nucleic acids with iodine will be delivered selectively to thyroid gland though iodine uptake.’

Round 2
Reviewer 1 Report
They try to answar all the qustaions.
Reviewer 2 Report
In the revised manuscript as well as the point-by-point responses, authors have properly addressed the concerns raised by the reviewer. Therefore, the reviewer thinks that the current manuscript has been sufficiently improved to warrant publication in IJMS.